# Using experience-based co-design (EBCD) to develop high-level design principles for a visual identification system for people with dementia in acute hospital ward settings

Alastair Macdonald  ,[1] Karolina Kuberska,[2] Naomi Stockley,[3] Bev Fitzsimons[3]

¹School of Design, The Glasgow School of Art, Glasgow, UK
²THIS Institute (The Healthcare Improvement Studies Institute), Department of Public Health and Primary Care, University of Cambridge, Cambridge, UK
³The Point of Care Foundation, London, UK

**Correspondence to**
Prof Alastair Macdonald;
A.Macdonald@gsa.ac.uk

## ABSTRACT

**Objectives** We tested a modified co-design process to develop a set of high-level design principles for visual identification systems (VIS) for hospitalised people with dementia.

**Design** We designed and ran remote workshops in three phases with carers of people with dementia and healthcare staff. In phase 1 we presented participants with scenarios based on findings from prior research, prompting participants to discuss their own experiences of VIS. Phase 2 used more future-focused scenarios, prompting participants to co-design improved VIS. In phase 3, a set of provisional design principles developed from our analysis of phases 1 and 2 data were discussed.

**Setting** Online workshops.

**Participants** A total of 26 carers and 9 healthcare staff took part in a pilot and three separate workshops.

**Results** We identified a set of six dementia-friendly design principles for improving the effectiveness of VIS: (1) The hospital trust provides a professionally-trained workforce and an appropriate culture of care; (2) the symbol is easily recognisable and well understood; (3) key personal information is readily available and accessible; (4) key personal information is integrated into the electronic patient record; (5) relatives and carers are involved in providing key information and monitoring care; (6) the principles need to function as a system to be successful. Participants suggested that, in addition to the use of an identifier and key personal information, professional standards training, effective information and records management and improved means to involve carers and/or families were key to the effective operation of VIS, leading us to expand a narrow understanding of a VIS.

**Conclusion** Using a scenario-led co-design approach can help trigger useful discussions with staff and carer groups, identify current problems with VIS and develop a set of high-level design principles for their improvement. These principles reveal day-to-day frictions that require further attention and resolution.

## INTRODUCTION

People with dementia and some other forms of cognitive impairment in hospital settings face a range of risks that present important

## STRENGTHS AND LIMITATIONS OF THIS STUDY

⇒ The experience-based co-design (EBCD) approach is adaptable and can be used to promote discussion of abstract high-level service principles as well as identify practical improvements to service design and delivery.

⇒ Designing and running a programme of online workshops using scenario-based narratives is a feasible and accessible alternative to holding in-person workshops.

⇒ The EBCD structure helps bonding between researchers and participants, in turn allowing for insightful and richer discussions, and for developing consensus.

⇒ The study would have benefited from a broader representation of healthcare professionals including, for example, ward managers, healthcare assistants, outpatients' department staff and peripatetic staff such as porters, as well as people with dementia and their carers.

challenges for care. One possible intervention to support care is what is commonly referred to as a visual identification system (VIS). This can help staff recognise people with cognitive impairment quickly and easily, and adapt their care accordingly. VIS have normally been understood as having two key elements, an 'identifier' and 'information'. The identifier is usually an easily noticeable sign or symbol, such as a specially coloured wristband, flower or butterfly.[1 2] The presence of the identifier signifies to staff that further important information about the patient's care and personal needs should be consulted and appropriate action taken; this additional information can be made readily available via patient profile documents, for example, a patient passport,[3] 'This is me' booklet,[4] 'About Me' records[5] or bedside poster. The Royal College of Psychiatrists recommends

that hospitals have a system in place to ensure staff are aware of a patient's cognitive impairment.[6 7] Their 2019 report stated that most UK hospitals are using some type of system.[7] However, the college does not recommend a specific form such a system should take, recognising that existing systems are implemented locally and they are likely to vary in cost and effectiveness.[7 8] Although VIS are routinely used within the National Health Service (NHS), some issues have been identified.[9] A number of localised versions of VIS are in use, but none has been rigorously evaluated or subject to systematic development informed by dementia-friendly design principles.

The DA VINCI (Developing A Visual IdeNtification method for people with Cognitive Impairment in institutional settings) programme looked at the use of VIS for people with dementia in hospital settings, with a view to identifying an existing system that might be further developed or, alternatively, co-designing a new system around the needs and preferences of patients, carers and health professionals to ensure acceptability and workability in practice.[8] DA VINCI took a multidisciplinary approach involving four linked studies from different organisations. These comprised: a survey of current practice around the visual identification of people with cognitive impairment in hospital settings[6]; an analysis of the ethical and legal issues involved in the use of such systems[10]; a qualitative study, including in-depth case studies of and interviews about current applications of VIS in hospitals with staff, people with dementia and their carers[11]; and a co-design study comprising a series of participatory workshops with carers and staff to co-develop a set of design principles. This paper describes the co-design study, the high-level principles identified and how these were produced.

### Study question and objective

The earlier DA VINCI survey and qualitative study identified various issues due to the proliferation of systems, suggesting that developing another VIS may be the wrong approach. We decided to explore the possibility of using an established co-design approach to identify a set of high-level principles to guide the development and use of any VIS, both existing and those developed in the future. Our objective was to co-develop and evaluate these high-level principles with healthcare professionals and carers of people with dementia.

### Co-design

Co-design is a collaborative process in which multiple stakeholders have input. It evolves as a process, maturing and adapting as it takes place. Co-design also involves a transformation of ordinary power relations between stakeholders and aims to generate collective ownership.[12] Experience-based co-design (EBCD) is a structured approach that has been developed to enable 'staff and patients (or other service users) to co-design services and/ or care pathways, together in partnership'.[13] EBCD has been used to achieve quality improvements in healthcare settings by bringing staff and patients together through a six-stage (or accelerated five-stage) process structured within a single study.[14 15] Examples include improvements to a mental health inpatient service and a pretreatment care pathway for patients with cancer.[16 17] Normally, in the early 'discovery' phase of EBCD, trigger films (videoed witness accounts of particular issues relating to the service in question) are created to prompt discussion at separate, and then joint, staff and patient events.[18] Staff and patient groups then come together during a later 'co-design' stage to suggest improvements to healthcare delivery and experience. We modified the EBCD approach given the separate study set-up in the DA VINCI programme, relying on the findings of the qualitative study[11] for our discovery phase. Also, due to the pandemic, we were unable to create trigger films to prompt discussion. Instead, we developed a set of scenario-based narratives relying on the findings of the qualitative study and prior experiences of our study team. We ran the workshops online.

## METHODS

### Materials: scenario-based narratives

We prepared a series of text documents describing short scenarios facing a fictional patient with dementia (Alice) admitted to hospital. These highlighted several dementia-related issues particular to Alice at a hospital-level, ward-level and patient-level, interspersed with key questions around issues that potentially affected Alice's care where a VIS might have an important role to play. Two further sets of scenarios were similarly developed, for patient George with Lewy bodies dementia and ward manager Claire. All three were translated into an illustrated slideshow format with added actor voice-overs. We showed some slides with the actor voice-over, paused and asked questions (see figures 1 and 2, and online supplemental file 1) to prompt a discussion about the highlighted issues and then moved on to the next slides and questions. Given the variety of VIS in use in UK hospitals, the scenarios and questions were intentionally generic rather than specific to any single VIS and were designed to support a structured workshop of about 2 hours' duration. Workshops were run by BF and NS, professional facilitators experienced in the application of the EBCD approach within healthcare through the Point of Care Foundation (PoCF). AM, a designer working with healthcare professionals using design-led co-development methods, assisted with note taking and some phase 2 facilitation.

### Study design

The three scenarios with voice-overs described above were shared with DA VINCI's Expert Collaborative Group (ECG) for comment and then piloted in a workshop involving five carers (box 1). There, participants found it difficult to shift from preoccupations with their experiences of existing VIS to think more speculatively about improvements. We therefore decided to hold the workshops in two separate phases a week apart. Phase 1 focused on identifying critical issues with the status

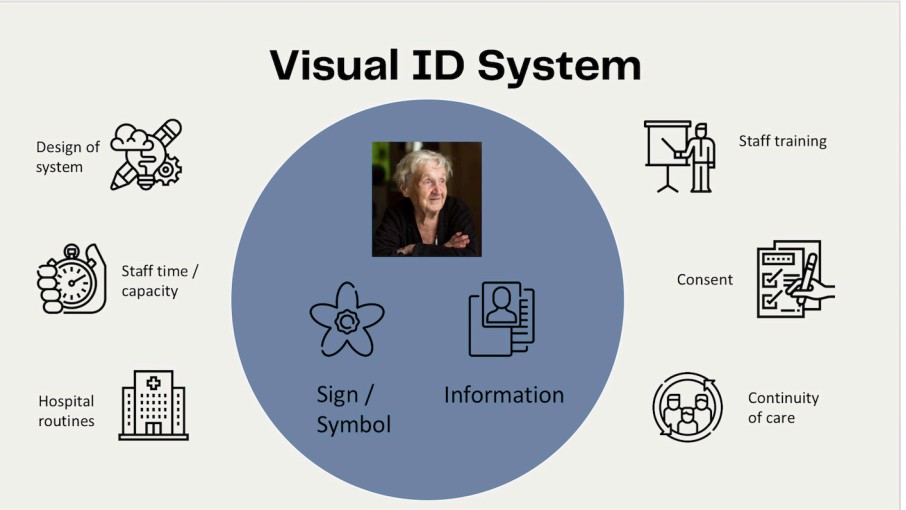

**Figure 1** The introductory slide from the stage 1 workshop to explain the concept of the visual identification system with its two key elements (within the blue circle) and how these relate to other relevant aspects of the hospital care environment.

quo; phase 2 allowed for more speculative thinking to improve these.[19] The phase 1 scenarios covered: (i) the frail older patient Alice, her VIS, being in hospital and issues of consent for the use of the VIS; (ii) patient George, and his care while being moved around the hospital; and (iii) ward manager Claire, responsible for ensuring staff consulted relevant information and provided person-centred care. The phase 2 scenarios focused on improving three specific aspects: (i) the identifier; (ii) the information; and (iii) how the VIS could function better. Phase 1 and phase 2 workshops were held with carers and staff separately; all participants were sent a copy of the slides prior to these workshops for their consideration. Following analysis of data from these two phases, we drafted a set of provisional design principles. These were discussed in the phase 3 joint workshop with carers and staff. This reviewed set of principles and feedback were then brought to a meeting of the ECG for discussion and consolidated after addressing their feedback. Box 1 summarises the study design.

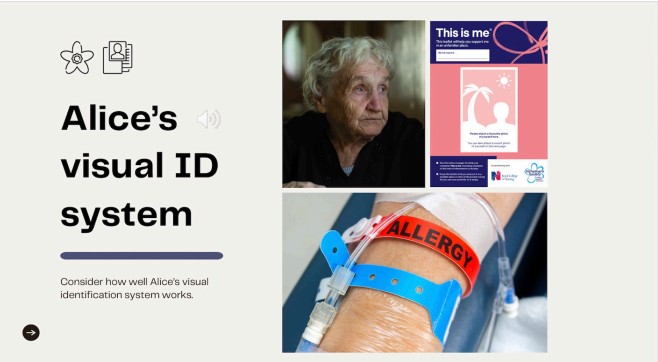

**Figure 2** A further slide from the stage 1 workshop illustrating key components of the visual information system for a fictional patient 'Alice', that is, the 'visual identifier' and the 'information element'.

### Recruitment

On behalf of the PoCF, the charity Dementia Carers Count circulated an invitation to take part in the workshops to their members. Interested participants had to contact the PoCF. There was a range of expertise among the carers who participated: length of time caring; type and severity of the dementia being cared for; setting in which care is/was provided; and experiences of hospital care, and the variability of hospital routines. Healthcare staff were recruited through a call on social media from the PoCF. Everyone who responded was invited to take part. All carers who responded and participated were women. Healthcare staff participating represented a range of expertise including old age psychiatry and specialist dementia nursing; all but one were women. Box 1 provides participant numbers at each stage. All workshop participants were offered compensation for their time in line with the INVOLVE (the UK's public participation charity) guidelines, which set out recommended rates of reimbursement for public participation in research.[20] For the purposes of this study, all participants (whether staff or carer) were treated as public participants in research and were compensated accordingly.

### Consideration of appropriate workshop tools for participants

As we were keen to remove barriers to participation, make the workshops as inclusive and as accessible as possible and so lessen the cognitive load, we decided against the use of any other tools than a proprietary online conferencing tool (Zoom). Many people were familiar with this tool by the time the workshops were held (autumn 2021), whereas many would not be familiar with other remote participative tools, such as collaborative online whiteboards.

### Data collection and initial analysis

The qualitative analysis was informed by Ramanadhan *et al*'s pragmatic theoretical framework, focusing on

**Box 1    Study design, showing the position of the three workshop phases within the study design, participant numbers in each, the materials used and nature of the questions explored. Online supplemental material provides details of the structure and questions for each of the three workshops**

Study Design proposal involving the DA VINCI partners and the Expert Collaborative Group

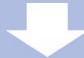

Development of workshop materials, piloting the 'status quo' scenarios with carers (n=5).

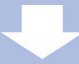

Phase 1: separate workshops for staff (n=2) and carers (n=7) using the 'status quo' scenarios exploring issues with existing visual information systems.
Phase 2: separate workshops for staff (n=3) and carers (n=7) using the 'what if?' scenarios exploring improvements to visual information systems.

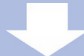

Analysis of data from phase 1 and 2 workshops, leading to a provisional set of design principles discussed with DA VINCI (Developing A Visual IdeNtification method for people with Cognitive Impairment in institutional settings) partners.

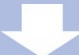

Phase 3: joint workshop with staff (n=4) and carers (n=7) exploring the concept of, viability of and issues with the provisional set of design principles.

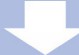

Analysis of feedback from phase 3 workshop, incorporating recommendations of the Expert Collaborative Group, to confirm our findings.

obtaining a practical understanding of real-world issues.[21] Their paper offers guidance on pragmatic approaches to analysis, which informed the approach in this study. They define the pragmatic approach as 'strategic combining and borrowing from established qualitative approaches to meet the needs of a given study'.[21] This approach led us to decide explicitly on an inductive process to generate an interpretation primarily grounded in and driven by the data (rather than existing theory). We privileged the perspectives of participants rather than ourselves as researchers and we intentionally connected the analysis plan to the outputs needed, that is, the design principles. Researchers AM, BF and NS gathered field notes during all workshops including, wherever possible, verbatim remarks. Each researcher independently coded their field notes and identified the themes around challenges and benefits of using VIS. These were then compared, and a consensus on the key themes was reached between the researchers in a series of online meetings. All workshops were video recorded, and if there were areas of lack of clarity or agreement about the key themes, the recordings were consulted to help resolve the issue. We analysed our field notes focusing on recurrent themes and patterns of experiences from field notes or paraphrasing common ideas from the conversations that took place in workshops. We describe the process of drafting and refining the co-design principles below.

### Patient and public involvement
No patients were involved in the design, conduct, reporting or dissemination plans of this study. Due to the pandemic, workshop participants did not include people with dementia, but any future studies would involve, with appropriate support, people with dementia and their carers as their experience should be key to nuancing the design principles of a VIS. The study teams were assisted by the ECG of the DA VINCI programme composed of healthcare professionals working with patients with dementia, carers for people with dementia, and third sector representatives, who were invited to comment on the methodology and findings of the studies throughout the project. Our initial study plans were shared with the ECG at the outset, the detailed methodology was shared mid-way through the project, ahead of the delivery of the workshops and the ECG were invited to comment on our first draft design principles. The results of the study have been disseminated to all staff and carer participants, partner study teams and the ECG.

### Ethical considerations
The project followed ethical procedures with written information provided in formats suitable for the target participants. The methods and activities undertaken in these workshops were chosen with the needs of people with dementia and their families in mind. Given the restrictions due to the COVID pandemic, workshops were organised and delivered remotely. Facilitation was undertaken by researchers familiar with the co-design approach and with experience of working with individuals who might be vulnerable, due to the personal nature of the subject of discussion. Researchers reassured participants that there were alternative ways to contribute if they were not comfortable using a computer or tablet, for example, offering to send a printed version of workshop materials and asking for comments in whatever way was most comfortable for them. Consent was gained to record the online workshops and to use anonymised quotes, and images of design ideas in our report. No participant was identifiable in any reports to the funder or in other

Table 1    Themes, subthemes and key points identified from workshop analysis

| Themes, subthemes and key points arising from workshop analysis | | |
|---|---|---|
| **Theme** | **Subtheme** | **Key points identified from workshop notes** |
| Training in dementia care | Lack of dementia expertise. Lack of awareness of relevance of the symbol. | Experiential training helps staff really understand what it's like to have dementia.<br>The UK Core Skills Training Framework includes dementia awareness training for every member of staff. Tier 2 training is required for every patient-facing member of staff.<br>Training sessions need to be more frequent; many staff members do not have up-to-date training. |
| The identifier and links to information | Lack of consistency between hospitals. No link to data. | Information about diagnosis needs to link to alert on the electronic patient records.<br>Linking primary care records with hospital records such that if a person with dementia presents to hospital, specialist dementia service is notified.<br>Critical information should be presented succinctly (not 4–5 pages). |
| Information | Information should be relevant, succinct and easy to access. | Include key pieces of information about the person (preferences for name, physical contact, food, lasting power of attorney, do not resuscitate (DNR), etc) and ways to contact family and carers.<br>Patient profile documents would benefit from a redesign towards presenting key information in a succinct way. |
| Family participation | Input of family is vital. Involving family to establish patient preferences and legal provisions. | Routinely contacting family and carers, especially when contact with patient is difficult to avoid making assumptions about legal provisions (eg, DNR orders). |
| Standardisation | Ensuring standard of care. | Expectation that patients receive similar levels of care irrespective of which hospital they stay at. |

outputs from the study. We checked regularly whether people were happy to continue and emphasised that they could withdraw from the workshops at any time. We also conducted a follow-up evaluation of workshops, so we could understand the experience of people taking part.

## RESULTS

We coalesced recurrent subthemes into distinct, identifiable themes, highlighting the key issues. Examples of the key themes identified from the coalescence of points arising from our workshop notes are shown and summarised in table 1.

### Training

Participants often mentioned the need for regular professional standards training. Of particular concern was high staff turnover and the challenges with ensuring consistency of understanding and communication, including using positive language when discussing patients with dementia, across both regular and bank staff. The need for training also applied to other non-clinical staff, such as porters. Carers emphasised the importance of institutional leadership and organisational culture. They also highlighted that experiential training sessions can be more effective for staff, as healthcare professionals are more likely to retain the information if they are learning by doing. There was consensus that the training needed to be of sufficient quality—recognising that mandatory training, especially online, can be of poor quality.

### Linking the identifier to key personal information

Participants pointed out the problems arising from an individual's key personal information not accompanying them around various healthcare settings. Some saw the solution in technology, such as smart-linking key personal information to the identifier (eg, through a QR or barcode), to make it readily available at each point in the care pathway. Some participants saw value in relevant summary information about the patient also being available in a physical form, suggesting that there may be situations where staff might not have time or opportunity for an extra step of scanning a code or checking the electronic patient record.

### Integration and management of records

Participants often mentioned that key medical and personal information often exists in separate electronic and paper-based systems which are not integrated. Participants felt that VIS could be instrumental to much better information integration and management along the whole patient care pathway, to include key search functions, automated alerts and links to that individual's care support needs. Again, many suggested that technology may offer useful solutions, from wearables to individual devices with screens at patient bedside.

### Enabling carers and families to provide key information

Participants highlighted the need for up-to-date, readily available, accessible and prioritised key information about each hospitalised individual with dementia, for example, preferred name, food preferences, allergies, ability to communicate and/or give consent, lasting power of attorney, and 'do-not-resuscitate' orders, to enable appropriate, personalised care. Relatives and carers felt, where possible, they should be enabled to input and update key personal information and emphasised the importance of being able to connect to their family member via a video call.

### Standardising the identifier

Perhaps the most contentious issue to arise relating to the identifier was what participants referred to as 'standardisation'. Many workshop participants considered that a standardised, nationally mandated identifier, applied on admission with assumed consent (unless revoked), might provide greater consistency in recognising patients with a confirmed diagnosis of dementia across different healthcare organisations, particularly given staff mobility across different sites using different systems. However, workshop participants who were healthcare professionals had concerns about how difficult it would be to implement a unified system across the whole NHS given that different VIS are already in existence.

### DESIGN PRINCIPLES

Plsek et al state that '[d]esign rules are heuristic statements in the form: If you want to achieve outcome Y in situation S, something like X might help'.[22] Our approach to deriving design principles (using the term 'principles' in preference to 'rules') from the phase 1 and 2 thematic analysis was adapted from their concept, which 'helps convert the tacit knowledge of organizational change agents into explicit, actionable knowledge'.[22] We used an adapted version of their methods 3 ('listen to stories of change efforts told by change leaders, operational managers, and front-line staff and then extract design rules off-line' (via review of transcripts or notes)) and 4 ('pose hypothetical scenarios to those experienced in organizational change, ask them to "think aloud" regarding how they would approach the situation, and then extract design rules off-line') to extract from our findings, through a number of iterations, a set of five draft principles for the joint phase 3 workshop. We provided these in concise form to all participants in a single sheet, which we supplemented with additional explanations at the workshop with further expansion on the principles. We invited participants to comment on individual principles in small breakout groups and at plenary sessions. We asked: 'If these principles were adopted, would: hospital staff recognise people with dementia more quickly; respond to their needs more readily; and provide better, more personalised care?' BF, NS and AM took notes of the discussion.

In the phase 3 workshop and in the feedback from the members of the ECG, participants generally agreed that some of the principles would be more problematic to implement than others, for example, improved information technology systems where the various clinical records, such as from hospital, general practitioner, care home and memory clinical records, could be better integrated with readily accessible key personal information. The point was also made that relying purely on electronic records was risky and that supplementary paper-based information would be essential. When asked which of the principles would be more fundamentally important to implement, phase 3 participants debated two in particular: training (contingent on organisational culture and leadership) and the form of the identifier (a wristband, symbol etc).

Opinions were divided over which principle was the most important. Some participants thought that a national, standardised identifier would be relatively easy to implement, especially with third-sector support for dissemination. However, as pointed out by members of the ECG, given how well-established some systems already are, a single nationally mandated identifier would be difficult, if not impossible, to implement. With regards to training, inadequately trained staff and lack of an ongoing training programme for all patient-facing employees, particularly for new or agency staff, could lead to them not being able to interpret the identifier in a way that supports patients as individuals. On a number of occasions, training was referred to as the 'bedrock principle'.

A key point from both the phase 3 workshop and the subsequent ECG discussion highlighted the interdependency of the principles which, if adopted singularly and piecemeal, would be unlikely to achieve the goal of more consistent, personalised care for hospitalised people with dementia. In other words, these principles need to be implemented systemically and as a set, if their benefits are to be realised. As a result, we added the interdependency of principles as a further principle. The feedback we received from phase 3 participants, the ECG and the project partners helped refine further our principles. The challenge involved making the principles simple yet adaptable so that they can serve a large, varied population in the most effective way possible. We provide a summary of the six working principles we arrived at through the phase 3 workshop in box 2.

### DISCUSSION
#### Using an EBCD approach

Whereas there are many variations in the design of VIS, we believe this has been the first attempt to develop a set of high-level principles to guide the design and use of VIS, through a participative co-design approach. Our study shows that designing and running a programme of online workshops using scenario-based narratives is feasible and that these may reduce some of the time and cost while improving accessibility in comparison to

Box 2  The set of six design principles resulting from the study

Design principles
1. The hospital trust provides a professionally trained workforce and an appropriate culture of care.
2. The symbol is easily recognisable and well understood.
3. Key personal information is readily available and accessible.
4. Key personal information is integrated into the electronic patient record.
5. Relatives and carers are involved in providing key information and monitoring care.
6. The principles need to function as a system to be successful.

in-person workshops. Presenting narratives on slideshows with voice-overs can effectively replace videoed witness accounts typically used in EBCD studies. The EBCD approach can also be used in evaluation studies focused on abstract, high-level design, such as developing principles for VIS.

### Reconciling abstract principles with practical reality
In phases 1 and 2, we focused on the more tangible elements of the VIS—the identifier and the key information and how these might be made to work better, which in turn provoked responses about issues concerning the less visible elements, leading to our findings. We acknowledge that formulating our findings into principles using Plsek *et al*'s heuristic was challenging for us (requiring several iterations) and that these were not fully resolved in their initial form in the phase 3 workshop. None-the-less, these principles served as a means of engaging participants in a more abstract level of discussion, along the lines of 'If we were to propose a set of design principles such as these, what would be your response be?'. Participants readily engaged with this more abstract form of discussion.

The value of the phase 3 and ECG feedback to our draft principles was to reveal the 'sticky materiality of practical encounters',[23] that is, the potential gap between high-level principles and the friction-laden character of situations on the ground, as well as the need for designing around what happens in reality of how identifiers for people with dementia are used in practice. Therefore, the sixth principle, emphasising the need for the first five principles to be implemented simultaneously is crucial. Discussing draft principles also highlighted the importance of a participatory process, whereby a proposed improvement, in prototype form (in this case our draft principles), can be interrogated and iteratively refined.

### Broadening the understanding of an effective VIS
While the two elements—the identifier and some form of document holding key personal information—are undeniably important, it became clear that for a VIS to function effectively, other elements—as identified in our findings—play a vital role in a functioning of the system that benefits the patient with dementia. For instance, lack of training may mean that even though a symbol is used for a hospitalised patient with dementia, some staff members may not be able to interpret it as a prompt. Involving carers in providing key information and monitoring care can help ensure that staff are able to adjust their care to the needs of each patient (eg, using preferred name or assisting with meals) rather than to exclusively rely on their broad training.

### Strengths and limitations
The EBCD approach is adaptable and can be used to promote discussion of abstract high-level service principles as well as identify practical improvements to service design and delivery. Designing and running a programme of online workshops using scenario-based narratives is a feasible and accessible alternative to holding in-person workshops.

The EBCD structure helps bonding between researchers and participants, in turn allowing for insightful and richer discussions and for developing consensus. The study would have benefited from a broader representation of healthcare professionals including, for example, ward managers, healthcare assistants, outpatients' department staff and peripatetic staff such as porters, as well as people with dementia and their carers.

### CONCLUSION
Our study shows that an adapted co-design approach using scenarios can trigger useful discussions with staff and carer groups, identify problems with VIS and engage participants in an abstract discussion about high-level design principles; and how these might guide the improvement of VIS.[20] Equally importantly, there was a consensus that the efficiency of the principles relied on their interdependent functioning. If only some principles are adopted singularly and piecemeal, this could lead to inconsistent, less personalised care. Thus, these principles need to be implemented systemically and as a set, if their benefits are to be realised. While we recognise these principles are not yet fully resolved, we have made progress towards better defining these principles. Constraints on the normal EBCD approach imposed by the linked-study commissioning method and the pandemic proved an initial challenge; ultimately, they were not an impediment to creatively developing an effective approach. This method could potentially be applied to other forms of service evaluation and improvement initiatives.

**Acknowledgements**  We acknowledge the helpful feedback, at all stages in our study from Graham Martin, Director of Research at THIS Institute (The Healthcare Improvement Studies Institute), and at key points, from our Expert Collaborative Group. Actor Francesca Isherwood provided the voice-overs for the Phase 1 materials. Joanna Goodrich presented the study with NS to the HSRUK conference in Sheffield in July 2022. We gratefully acknowledge the participation of the carers and staff in the workshops, and members of the Expert Collaborative Group for their constructive input. We are grateful to the charity Dementia Carers Count for connecting us with the carers included in this study. The icons in figures 1 and 2 are used by kind permission of Flaticon.com.

**Contributors** BF, NS and AM were the researchers, led by BF, comprising the co-design study and all contributed to the drafting of this manuscript, led by AM. KK reviewed and substantially edited and contributed to the manuscript. All authors reviewed the final version. AM acted as guarantor.

**Funding** Health Foundation, Grant/Award Number: RHZF/001 - RG88620; Health Services and Delivery Research Programme, Grant/ Award Number: NF-SI-0617-10026.

**Competing interests** None declared.

**Patient and public involvement** No patients and/or the public were involved in the design, or conduct, or reporting, or dissemination plans of this research. Refer to the Methods section for further details.

**Patient consent for publication** Not applicable.

**Ethics approval** This was a service improvement study and as such it did not require an ethical approval. Participants gave informed consent to participate in the study before taking part.

**Provenance and peer review** Not commissioned; externally peer reviewed.

**Data availability statement** Data sharing not applicable as no data sets generated and/or analysed for this study.

**ORCID iD**
Alastair Macdonald http://orcid.org/0000-0001-9282-6229

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
