## [Reviewer comments · BMJ Open]

ARTICLE DETAILS

TITLE (PROVISIONAL)	Using experience-based co-design (EBCD) to develop high-level design principles for a visual identification system for people with dementia in acute hospital ward settings
AUTHORS	Macdonald, Alastair; Kuberska, Karolina; Stockley, Naomi; Fitzsimons, Bev

VERSION 1 – REVIEW

REVIEWER	Lillian Hung Vancouver General Hospital
REVIEW RETURNED	01-Feb-2023

GENERAL COMMENTS	Thanks for the opportunity to review this interesting paper, which provides design principles for visual identification systems (VIS) for hospitalized people with dementia. I have a few questions and suggestions for the authors. 1. Objectives and research questions were not clearly stated. To improve VIS? To co-develop design principles? What was the study question?2. How did the authors decide on data saturation/ sufficient data collected to answer study questions? Any follow-up interviews after the workshops?3. Sampling. Please provide information about the participants' demographic information for both staff and care partners. It would be nice to attach a table with such information.4. Describe the researchers and any design and clinical background. Did you have a UX professional facilitator to lead the workshops?5. Workshops: Provide relevant information about workshops. e.g. What tools did they use? (e.g. Miro?) What are the efforts made to ensure participants were meaningfully involved? What technical challenges did they help participants to overcome? What is the level of participation of each participant from different professional backgrounds? Lessons learned from these remote workshops can be so helpful in informing future studies.6. Discussion is so short and could be expanded. The discussion should connect to the introduction and the research questions. In your introduction, you talked about EBCD and VIS. Please describe the significance of your findings in relation to what was already known about the research problem being investigated and explain any new understanding or insights that emerged as a result of the remote workshops. Cite similar work to show how your study related to current literature. Explains how your study advanced the reader's understanding of the research problem.7. Implications. I am excited to see the work of online workshops using scenario-based narratives. Your photos demonstrate positive and inspiring examples. I really like the voice-over slides. Provide recommendations for future studies?
---

REVIEWER	Alethea Blackler Queensland University of Technology, CIESJ
REVIEW RETURNED	02-Feb-2023

GENERAL COMMENTS	This is a very interesting and important project, and highlights the contribution that disciplines like design can make in the health field. However, it is very important to improve clarity in the reporting of this research as readers of the BMJ are not generally subject experts in co-design and the lack of clarity in reporting this project will reduce the likelihood of it having the impact it deserves. My suggestions to remedy this are:  1. Include a clear research question, ethics statement (although in theory not needed, how did you inform and gain consent form participants? What exactly is the INVOLVE framework?), and limitations section. 2. Do not use confusing or obscure acronyms - e.g. EBCD will not be well recognised by the audience and ECG already means something very different to your potential audience. Just put these in full. 3. Define co-design more thoroughly for the non-cognate audience you are targeting. 4. On page 4 you mention the findings of your qualitative study - can you cite something here? 5. When you describe your pilot, please include more information - how many workshops did you pilot? With whom? 6. Table 1 describes your study design somewhat but I would rather have it spelled out clearly in written form alongside a clear flowchart or similar. 7. You mention a pragmatic theoretical framework but do not describe it beyond that. I suggest put an explanation or remove reference to it. 8. Explain how you collected and analysed your field notes. Did you use a proforma? Were you able to record in other ways such as video? What coding scheme did you use to analyse the notes and any other data? Is this what Table 2 is? Did you use any software? You need to make sure you explain the rigour of your research thoroughly for it to be taken seriously by this audience. 9. In table 2, what do LPA and DNA refer to? 10. Explain more clearly the methods 3 and 4 involving story-telling - again focus on communicating your rigour. 11. There is a lot of information submitted for supplementary material but not enough detail, examples and illustration in the text itself - can you find a balance? Not everyone will want to access to supplementary stuff. Supplementary should be supplementary - anything essential to understanding your project should be in the main text and figures. 12. You have developed principles but do not describe them on the text - just a list or image in line with the content and not in a supplementary file would be suitable. 13. Also you do not reiterate your refined principles after phase 3 workshop. 14. At the end of the paper, I am unclear what your findings and/or recommendations are. These should be made crystal clear. I think this work has a lot to contribute and would love to see it well represented and explained so that it can have impact. Thanks for asking me to review it.
--

VERSION 1 – AUTHOR RESPONSE

R1	Dr. Lillian Hung, Vancouver General Hospital, Simon Fraser University	
	Thanks for the opportunity to review this interesting paper, which provides design principles for visual identification systems (VIS) for hospitalized people with dementia. I have a few questions and suggestions for the authors.	
	Comment	Response
	1. Objectives and research questions were not clearly stated. To improve VIS? To co-develop design principles? What was the study question?	We have added a 'Study question and Objective' section into the Introduction.
	2. How did the authors decide on data saturation/ sufficient data collected to answer study questions? Any follow-up interviews after the workshops?	Thank you for this question. The COVID-19 pandemic was a factor limiting recruitment, particularly for staff participants. We indicate this staff numbers issue in the 'Strengths and limitations' section. However, the small numbers worked well. We made efforts to ensure that the workshops were a safe space for discussions of a sensitive nature and so trust was quickly developed between the participants and researchers and a good bond was established between participants themselves within the limited workshop time. This also allowed for intimate and richer discussions and led to a good degree of consensus. We discuss these issues in the 'Strengths and limitations' section.
	3. Sampling. Please provide information about the participants' demographic information for both staff and care partners. It would be nice to attach a table with such information.	The expanded 'Recruitment' section provides details of recruitment. We did not collect participants' demographic details as we were primarily interested in the richness of participants' lived experience and the degree of consensus around principles being developed rather than being representative of a particular population. However, all our respondents and participants were women apart from one and we have added this note into the text in the 'Recruitment' section and referred readers to Table 1 which contains the numbers of participants at each stage.
	4. Describe the researchers and any design and clinical background. Did you have a UX professional facilitator to lead the workshops?	We have added a short statement about the expertise of the researchers at the start of the 'Methods' section.

	5. Workshops: Provide relevant information about workshops. e.g. What tools did they use? (e.g. Miro?) What are the efforts made to ensure participants were meaningfully involved? What technical challenges did they help participants to overcome? What is the level of participation of each participant from different professional backgrounds? Lessons learned from these remote workshops can be so helpful in informing future studies.	We have added a short section under the heading 'Consideration of appropriate workshop tools for participants' where we explain and justify our approach.
	6. Discussion is so short and could be expanded. The discussion should connect to the introduction and the research questions. In your introduction, you talked about EBCD and VIS. Please describe the significance of your findings in relation to what was already known about the research problem being investigated and explain any new understanding or insights that emerged as a result of the remote workshops. Cite similar work to show how your study related to current literature. Explains how your study advanced the reader's understanding of the research problem.	We have added substantial additional material in new and existing sections in the paper. We have added a short piece about working with EBCD in the 'Discussion' section. We have also made a clear statement within the last paragraph of the 'Discussion' section about our claim to originality in that we believe we are the first to attempt to develop a set of high-level principles to guide the design and use of VIS, through a participative co-design approach.
	7. Implications. I am excited to see the work of online workshops using scenario-based narratives. Your photos demonstrate positive and inspiring examples. I really like the voice-over slides. Provide recommendations for future studies?	We have revised the 'Discussion' and 'Conclusion' sections to include recommendations.
R2	Dr. Alethea Blackler, Queensland University of Technology This is a very interesting and important project, and highlights the contribution that disciplines like design can make in the health field. However, it is very important to improve clarity in the reporting of this research as readers of the BMJ are not generally subject experts in co-design and the lack of clarity in reporting this project will reduce the likelihood of it having the impact it deserves. My suggestions to remedy this are:	
	1. Include a clear research question, ethics statement (although in theory not needed, how did you inform and gain consent from participants? What exactly is the INVOLVE framework?), and limitations section.	Thank you for this comment. We have included a section 'Study question and objective' in the introduction section, added 'Ethical considerations' section, have provided details of INVOLVE in the 'Recruitment' section, and provided a 'Strengths and limitations' section in the 'Discussion' section.
	2. Do not use confusing or obscure acronyms - e.g. EBCD will not be well recognised by the audience and ECG already means something	We have added the full definitions of both the EBCD and ECG acronyms at a number of points throughout the paper, allowing for

	very different to your potential audience. Just put these in full.	easier reference within sections where these are repeated. We hope that this approach will make it easier to follow our paper for readers.
	3. Define co-design more thoroughly for the non-cognate audience you are targeting.	Thank you for this comment. We have inserted additional detail about co-design and about experience-based co-design (EBCD) which would be useful to an audience not familiar with this cognate area.
	4. On page 4 you mention the findings of your qualitative study - can you cite something here?	We have added an additional reference (11): Sutton E, Armstrong N, Locock L, Conroy S, Tarrant C. Visual identifiers for people with dementia in hospitals: a qualitative study of a classification system for improving quality of care. BMJ Quality & Safety . forthcoming.
	5. When you describe your pilot, please include more information - how many workshops did you pilot? With whom?	Thank you for this comment. We have added more clarity in the section 'Study design'.
	6. Table 1 describes your study design somewhat but I would rather have it spelled out clearly in written form alongside a clear flowchart or similar.	We have added more detail to the 'Study design' section as well as revised Table 1 to make the information more concise and readable and show the flow of events.
	7. You mention a pragmatic theoretical framework but do not describe it beyond that. I suggest put an explanation or remove reference to it.	In 'Data collection and initial analysis' section, we have provided fuller details on Ramanadhan et al.'s pragmatic framework and the rationale for using this.
	8. Explain how you collected and analysed your field notes. Did you use a proforma? Were you able to record in other ways such as video? What coding scheme did you use to analyse the notes and any other data? Is this what Table 2 is? Did you use any software? You need to make sure you explain the rigour of your research thoroughly for it to be taken seriously by this audience.	We did not use a proforma. We state in the 'Data collection and initial analysis' section that we recorded all workshops on video which we consulted if there were areas of lack of clarity or to check details; these were also useful in detailing verbatim quotes. Table 2 shows the coding scheme themes and subthemes which emerged: the themes are each given a para each following Table 2. We hope the revised text gives more clarity.
	9. In table 2, what do LPA and DNA refer to?	Thank you for this comment. We have explained all acronyms in Table 2. We have corrected the DNA to DNR – do not resuscitate.

	10. Explain more clearly the methods 3 and 4 involving story-telling - again focus on communicating your rigour.	We have provided details of Plsek et al.'s methods 3 and 4 and removed the reference to storytelling as the fuller description of these methods now makes our method clearer.
	11. There is a lot of information submitted for supplementary material but not enough detail, examples and illustration in the text itself - can you find a balance? Not everyone will want to access to supplementary stuff. Supplementary should be supplementary - anything essential to understanding your project should be in the main text and figures.	This is a helpful suggestion. We have retained only the supplementary material comprising the summaries of the workshop structures and questions (now numbered supplementary materials 1) and another supplementary document for the editor only. We have now included two of the colour slides, which were previously supplementary materials, as figures within the main document. We removed the potentially confusing supplementary material 5 (interim principles used in Phase 3 workshop) in preference to a clearer summary of the six principles emerging from the Phase 3 workshop (Table 3).
	12. You have developed principles but do not describe them on the text - just a list or image in line with the content and not in a supplementary file would be suitable.	Thank you for this comment. We have provided a list of the six principles in Table 3.
	13. Also you do not reiterate your refined principles after phase 3 workshop.	As above, we have provided a list of the six principles in Table 3.
	14. At the end of the paper, I am unclear what your findings and/or recommendations are. These should be made crystal clear.	We have added a recommendations statement in the 'Conclusion' section.

VERSION 2 – REVIEW

REVIEWER	Lillian Hung Vancouver General Hospital
REVIEW RETURNED	11-Mar-2023
GENERAL COMMENTS	Thanks for your careful revision. I have no further comments.